# Path Forming of Healthcare Practitioners in an Indoor Space Using Mobile Crowdsensing

**DOI:** 10.3390/s22197546

**Published:** 2022-10-05

**Authors:** Brixx-John Panlaqui, Muztaba Fuad, Debzani Deb, Charles Mickle

**Affiliations:** 1Center for Applied Data Science (CADS), Winston-Salem State University, Winston-Salem, NC 27110, USA; 2Department of Computer Science, Winston-Salem State University, Winston-Salem, NC 27110, USA

**Keywords:** Bluetooth, RSSI, mobility, path forming

## Abstract

While there are numerous causes of waste in the healthcare system, some of this waste is associated with inefficiency. Among the proposed solutions to address inefficiency is clinic layout optimization. Such optimization depends on how operating resources and instruments are placed in the clinic, in what order they are accessed to attain a particular task, and the mobility of clinicians between different clinic rooms to accomplish different clinic tasks. Traditionally, such optimization research involves manual monitoring by human proctors, which is time consuming, erroneous, unproductive, and subjective. If mobility patterns in an indoor space can be determined automatically in real time, layout and operation-related optimization decisions based on these patterns can be implemented accurately and continuously in a timely fashion. This paper explores this application domain where precise localization is not required; however, the determination of mobility is essential on a real-time basis. Given that, this research explores how only mobile devices and their built-in Bluetooth received signal strength indicator (RSSI) can be used to determine such mobility. With a collection of stationary mobile devices, with their computational and networking capabilities and lack of energy requirements, the mobility of moving mobile devices was determined. The research methodology involves developing two new algorithms that use raw RSSI data to create visualizations of movements across different operational units identified by stationary nodes. Compared with similar approaches, this research showcases that the method presented in this paper is viable and can produce mobility patterns in indoor spaces that can be utilized further for data analysis and visualization.

## 1. Introduction

In the United States, 25% of healthcare spending is linked to waste, with estimates ranging from USD 760 billion to USD 935 billion in 2019 [1]. While there are many causes associated with waste, one significant cause is inefficiency. In pro bono clinics, this type of waste negatively impacts the quality of care provided because of factors such as limited resources and student clinicians providing services during their training. In North Carolina, many of these clinics are affiliated with many health systems and universities, with services being provided by volunteer clinicians and student practitioners. The second largest pro bono clinic in the United States, the Community Care Center (CCC), is located in Winston-Salem, providing services to patients from surrounding counties. 

The CCC is a multispecialty clinic that provides patients with a wide range of services. This clinic adopts a “spoke and wheel” configuration. The center of the “wheel” is the centralized charting station, and the patient rooms surround the station as the “spokes”. By knowing the general layout of the clinic, practitioners can move around knowing where they need to go. For physical therapy students, the goal for them in this scenario is to use the skills learned in the classroom within a real clinical environment. Due to clinic procedures and job demands, concern arises related to the optimization of patient care by streamlining processes. As students leave the room on a continuous basis for equipment and supervision, there is a concern of lost patient time. Strategies such as placing equipment closer to treatment rooms, streamlining the sequence of operations, assigning a supervisor to specific treatment rooms, and placing computers in the rooms to increase productivity have been attempted with varying success and effects on productivity. However, due to data inadequacy, it is not known what other strategies might be effective in increasing student physical therapists’ productivity to maximize time with the patient. To gain a better understanding of how best to optimize mobility, one valuable technology is mobile crowdsensing [2], which has attracted much attention because of its wide versatility across different problems and applications. Crowdsensing data and algorithms have the potential to automatically and adequately model the mobility around the clinic and therefore can enable researchers to better discover what clinic layout and operations are currently inefficient and how to optimize them. Bluetooth Low Energy (BLE) beacons are typically utilized for these types of localization problems, as they are easy to obtain. The signal strength of these beacons is measured in RSSI values. Yet, their precision, usability, and computational power are questionable regarding the localization of the data. 

Although the application domain of this research is clinic layout management, the scope of this paper is not. This paper demonstrates the technique for path forming, which is necessary to find patterns of mobility in the clinic. Once enough data are collected using the presented technique, patterns in mobility can be determined, and visualizations of such mobility can be produced for layout optimization. This study focuses on utilizing Bluetooth-enabled devices only and presents methods for finding a path of movement in an indoor space using Bluetooth RSSI data. These data consist of measurements of signal strength for Bluetooth connections between devices. Pinpointing precise locations is not within the scope of this research; instead, the goal is to estimate a general mobility path with respectable accuracy and find patterns of mobility in an indoor space. The research methodology involves developing two new algorithms that use raw RSSI data to create visualizations of movements across different clinical operational units identified by Bluetooth-enabled stationary nodes. The first path-forming algorithm requires knowing the coordinates of the indoor space (floorplan) and stationary devices to model the mobility. The second algorithm is designed to estimate the mobility without the knowledge of floorplans or the exact coordinates of stationary devices representing the various operational units within a clinic. 

In recent approaches to indoor localization [3,4], there have been promising results in pinpointing exact locations using Bluetooth. However, some approaches have shown the need to know the floorplan or dimensions of an area [5,6]. They rely on beacons [7,8,9], which are not always accessible and wield less computational power. Other works in healthcare have shown growing application in mobile crowdsensing, such as in clinical and psychological trials [10,11,12,13,14], public health [15], and personal well-being [16]. These works only focus on patient data rather than provider data. Although these techniques have some strengths and shortcomings, this research has different needs, goals, and approaches. This application domain requires the easy application of this technology without sophisticated hardware or the need for precise accuracy or efficient energy usage. Since mobility detection is the goal, the approach does not need high precision. Additionally, the use of stationary devices connected to power gives this approach the flexibility needed to use the Bluetooth proximity detection [17] technique in real time. With the incorporation of computationally intensive and power-savvy stationary mobile devices instead of specialized beacons with less computational power or network capability, this paper presents an approach to indoor mobility detection by using only mobile devices and gathered Bluetooth RSSI values. Table 1 lists the similarities and differences of this research with similar research works.

The use of smartphones allows important data to be quickly gathered and compared with other crowd users, especially when taking environmental factors, such as sudden fall and gait, into account alongside mobility. This unique ability highlights an opportunity to utilize mobile crowdsensing to collect provider data with the goal of improving both patient and provider experiences. 

## 2. Materials and Methods

This section starts by describing the background of this research, including data collection, experimental setup, and issues related to estimating the path of stationary and moving mobile devices. The section then gives a detailed overview of the two path-forming algorithms. 

### 2.1. Data and Data Collection Methodology

The datasets used in this study were created based on the raw data with various identifiers and the RSSI values (in dBm) that were collected during the above two experiments. After the experiments, the RSSI data from each stationary device were exported into either a .csv (for Algorithm 1) or plain-text (Algorithm 2) file with the following attributes:1.Timestamp (milliseconds);2.Date/time;3.Stationary device coordinates;4.Stationary device name;5.RSSI (dBm).

In both path-forming algorithms, these datasets are eventually curated and aggregated (Section 2.3.2 and Section 2.4.2).

To collect practitioners’ movement data, two different methods were conducted on the first floor of the New Science Building at Winston-Salem State University. For both experiments, three stationary devices were placed in specific locations across the indoor space representing various operational parts of the clinic. The mobile device that the practitioner carries is discoverable and always transmits a radio signal. This signal is detected by the stationary devices, and each stationary device has a mobile application that tracks the RSSI from the mobile device over time.

The mobile application, Bluetooth Scanner [18], finds all Bluetooth devices, including connected, paired, and unknown devices. It scans for all nearby discoverable devices with Bluetooth. After scanning, all discoverable devices are listed, as shown in Figure 1a. A device can be picked among those in the list by tapping “Device Finder”. This leads to a screen (Figure 1b) that shows the current signal strength from the chosen device. The RSSI value is displayed in dBm and will change over time. Below the signal strength, a graph displays the changes and fluctuations in the RSSI since selecting the chosen device. At any point, the RSSI data can be exported for further processing.

The first data collection method (Section 2.1.1) involves gathering RSSI data from the mobile device while it is stationary. There are 15 locations marked within this indoor space, and the mobile device is placed and remains idle at each of these fixed locations, while the stationary devices gather RSSI data for a certain length of time. The second data collection method (Section 2.1.2) involves the mobile device moving in real time and following specific paths around the indoor space. Similar to the first method, the stationary devices gather RSSI data from the mobile device. The following sections cover the two methods of collecting mobility datasets in detail.

#### 2.1.1. Estimating the Location of a Stationary Mobile Device

The purpose of the first data collection method is to collect RSSI data for the first path-estimation algorithm in Section 2.3, which required the estimation of single locations as part of its approach. The goal is to explore the accuracy and feasibility of using the measured power of the stationary devices in estimating the location of a mobile device when it is fixed. The measured power is a constant that indicates the expected RSSI at a distance of one meter from another Bluetooth device. The first step is to obtain the measured power between the mobile devices. To achieve this, two of the mobile devices are placed a meter apart, and the RSSI between the two is collected over a period of 30–40 s.

The second step is to assign locations to the stationary devices and measure coordinates. To start, the coordinates of the indoor space were obtained in inches, the three stationary devices (marked with blue) were placed around the room, and 15 fixed locations (marked with orange) across the room were also identified, as shown in Figure 2. This space was selected for data collection as it mimicked a space with traditional furnishings and obstacles that might have similar effects on the Bluetooth reception of our application domain. The coordinates of the three stationary devices and 15 locations were measured. These coordinates are only needed for the first of two path-forming algorithms, as discussed in Section 2.3. The other proposed algorithm is designed to estimate the path without knowing the coordinates of the indoor space and stationary devices.

Once the coordinates were recorded, the next step was to gather the RSSI data from all stationary devices while keeping the mobile device fixed in each of these 15 marked locations, as shown in Figure 2. The three stationary devices, S1 to S3, begin scanning until they find the mobile device. When the mobile device is located, all stationary devices will immediately scan it and record the signal strength values. After 30–40 s, the location data gathered from each stationary device are exported for further analysis.

#### 2.1.2. Estimating the Real-Time Movement of a Mobile Device

The purpose of the second data collection method is to gather location data while simulating different paths with the mobile device inside the indoor space. This involves collecting the RSSI as the volunteer walks with the mobile device in hand.

This method entails three different real-time mobility paths of the mobile device, as shown in Figure 3. Figure 3a shows Path 1, where the volunteer passes the mobile device through the nodes, in the following order: S1 to S3, S3 to S2, and S2 back to S1. In Figure 3b, Path 2 consists of the mobile device moving in a straight line from the left to the right side of the indoor space while being closer (sequentially) to stationary device S1, then S2, and then S3. Path 3 (Figure 3c) involves the mobile device starting to visit S1, then to S2, then S2 to S3, and finally from S3 back to S1. For Paths 1 and 3, the volunteer holding the mobile device travels the path and stops by each stationary device for a few extra seconds to mimic the real clinic operations. As the mobile device is moving, the stationary devices collected the RSSI from the mobile device using the Bluetooth Scanner application. All stationary devices export the data after the mobile device completes its movement throughout the specified paths.

### 2.2. Path-Forming Algorithms

To understand how mobility can be optimized in an indoor space, it is important to develop methods of identifying movement that maximize the use of signal strength data from Bluetooth. Due to the unpredictable nature of RSSI data, the ability to pinpoint exact locations could potentially be impeded. However, the goal of estimating a path is still very feasible. 

Two path-forming algorithms were developed and used to estimate the paths of mobility from the RSSI data. After generating their visualizations, these two algorithms are compared through their accuracy and ability to generalize. The first algorithm (Section 2.3) utilizes a trilateration-based approach in identifying the individual locations of a mobile device and combining them to visualize a path of mobility. The second algorithm (Section 2.4) makes use of vectors to form a sequence of stationary device proximities indicating a path. A detailed explanation of each algorithm is covered in the following sections.

### 2.3. Trilateration-Based Path-Forming Algorithm (Algorithm 1)

The first algorithm consists of converting the RSSI values captured at each of the three stationary devices into a radius around them. The intersection of the three radiuses, or the area of trilateration, is used to estimate the location. The center of the trilateration area is the estimated location of the moving device, and all locations are continuously incorporated into a single visualization that presents a path of movement. This algorithm works with data from both experiments (Section 2.1.1 and Section 2.1.2). However, this algorithm requires the indoor space dimensions, coordinates of the stationary devices and numbered locations as part of the input. The algorithm comprises three steps, given below and detailed in later subsections.
Aggregating data;Visualizing data;Estimating the path.

The algorithm utilizes Jupyter Notebook as the primary development application. Python packages such as pandas and matplotlib help to simplify data manipulation tasks [17], thus playing a big role in the decision to use Jupyter and Python to implement this algorithm.

#### 2.3.1. Aggregating Data

This step involves finding the median for both the measured power and the collected RSSI of the stationary devices at each marked location. As mentioned previously, the measured power is a constant indicating the expected RSSI when devices are a meter apart (Section 2.1.1). However, multiple RSSI values were gathered from the measured power, so only a single value is needed to represent the constant. For the collected data, all three stationary devices have their own list of RSSI values for each of the 15 locations, giving 15 datasets per stationary device. The median was chosen as the measure of tendency because of the potential for clear outliers. 

By taking the median of the collected data, each stationary device should have a single RSSI value for each marked location. To organize the median values based on a stationary device, the values should be divided into lists of *n*, which are all converted to a Pandas DataFrame. The following algorithm accomplishes these functions.
Calculate median from RSSI values in the measured power data.For each set of RSSI data per stationary device:
a.Calculate median of the set of RSSI.Calculate size of chunks *n* using the number of median RSSI divided by the number of stationary devices.For every *n* number of median RSSI:
a.Add to a list designated for the stationary device.Convert all lists of RSSI median values into a single DataFrame (table).

Once the raw datasets are combined, the aggregated datasets for Algorithm 1 consist of:1.RSSI (dBm) for S1;2.RSSI (dBm) for S2;3.RSSI (dBm) for S3.

#### 2.3.2. Visualizing Data

After the RSSI DataFrame is created, the next step is to visualize the data. The path-forming function performs multiple function calls for smaller algorithms of the different processes in visualization. The input for this function is:The RSSI values for each stationary device;Number of stationary devices;Indoor space dimensions;The x and y coordinates of the stationary devices and marked locations;Measured power of mobile devices;A constant “N” (Range 2–4, strength of the device).

When the path-forming function is executed, a loop iterates through each row of the DataFrame, where the rows represent the marked locations in ascending order. The first function called in the loop visualizes the data. 

In the path-forming function, a plot is made with the specified x and y dimensions. The locations of the stationary devices are plotted along with one of the 15 marked locations based on index. After preparing the plot, the RSSI values for each device in the row are retrieved and used to calculate distance. The distance equation relies on the use of the measured power and the constant “N” [19]. The equation for calculating distance [20] is shown below.
(1)$$\mathrm{Distance}=10^{\frac{Measured\;Power-RSSI}{10*N}}$$

The measured power is one of the constants used for this equation. The second constant, “N”, is the broadcasting power value of the device and ranges from 2 to 4 dBm (low to high). Due to the constraints of RSSI data, this does not give the exact distance from the mobile device. However, the approximate distances for the devices are used as circles, whose radiuses work in tandem to obtain an approximate location. The algorithm finds the distance and plots it as a radius, with the location of the stationary device as the center. The following function returns the plot and list of radiuses.
Create a plot with x and y dimensions.Plot x and y coordinates of stationary devices and marked locations.For each stationary device:
(a)Retrieve RSSI value from DataFrame at the current index, column;(b)Calculate *distance* using Equation (1);(c)Plot a circle with *distance* as radius and the stationary device as center;(d)Append radius of the circle to the list of radiuses.Return plot and list of radiuses.

#### 2.3.3. Estimating the Path

After visualizing the data, the next step is to estimate the device’s path. The below function call utilizes the *createList* function, which returns all combinations where two circles intersect. The input for this function is the number of stationary devices. All unique combinations of the list are identified and grouped into a single list, which is returned with the original list of indexes.
For each stationary device:
a.Append the index of the device to the list of indexes.Find all unique combinations among stationary devices.Return the list of combinations and list of stationary devices.

The third function call invokes the function *get_points_of_intersection_area*, which identifies the points surrounding the intersection area. The input consists of the combinations and the list of indexes from the previous function, the x and y coordinates of the stationary devices, and the stationary devices’ radiuses. The function creates two empty lists for the x and y coordinates. A loop iterates through the list of combinations and finds the missing index (circle). The second step in the loop is a function call that calls the *get_intersections* function.
Create two empty lists for x and y coordinates of the trilateration area.For each combination:
a.Find the stationary device that is not in the combination;b.Call the *get_intersections* function.

The *get_intersections* function above finds the intersection points for each combination of circles. This function’s input includes the x and y coordinates (x1, x2, y1, y2) and the radiuses of the pairs of intersecting circles. If the two circles intersect, the output of the function is a tuple with the x and y coordinates of their intersection points.
Calculate *D* as the square root of (x2 − x1)^2^ + (y2 − y1)^2^.If *D* is less than radius 1 + radius 2:
a.Return None.If *D* is less than the absolute value of radius 1 − radius 2:a.Return None.If *D* is equal to radius 1 and radius 1 is equal to radius 2:a.Return None.Else:Return coordinates of the two intersection points.

Returning to the previous function, *get_points_of_intersection_area*, there is a condition that checks if the *get_intersections* function returned the coordinates of the intersection points. If coordinates are returned, another check determines if the intersection points are located in the third circle’s radius. If true, the x and y coordinates are appended to the empty lists from earlier. The resulting output is the coordinates of points surrounding the intersection area.
For each combination (cont.):a.If *get_intersections* function did not return None:i.If Coordinates of all intersections are returned.b.For each set of coordinates:i.If coordinates are inside the third circle’s radius:Plot coordinates on graph;Append x and y coordinates to the lists of coordinates.Return lists of x and y coordinates of points surrounding trilateration area.

In the path-forming function, the lists of x and y coordinates for the intersections are returned. Now that all function calls are through, a check is performed inside the main function to determine if the length of both lists is greater than 2. This means there is a common intersection across all circles. If this condition is true, the averages of both lists are calculated. These averages become the estimated location.

If two or more pairs of coordinates surrounding trilateration area exist:a.Calculate the average of all pairs of coordinates of the trilateration area;b.Plot average.

After estimating the locations, the locations are combined to form a path. In the final step of visualizing the data, a plot is created to combine estimated locations from various timestamps and to form a path. In the final output, the predicted locations are connected with arrows indicating the direction and flow of movement.
Create a new plot.Plot locations of stationary devices and predicted locationsa.Connect predicted locations with lines and arrows indicating direction;Plot legend.

### 2.4. Vector-Based Path-Forming Algorithm (Algorithm 2)

The second algorithm involves sorting the RSSI data by time series and creating directional vectors to show the source and destination. Combined, these vectors show a flow of movement relative to the three stationary devices. Unlike the previous algorithm, the second algorithm does not require any indoor space dimensions or coordinates. 

The second algorithm was developed using Java [21]. Java itself is a popular programming language that is platform-independent, easy to learn, secure, and robust. The primary development environment for this algorithm is NetBeans, which is a free and open-source integrated development environment (IDE) for application development. 

This algorithm consists of four primary steps as below with their own sub-algorithms. These steps are detailed in the following subsections:Aggregating data;Combining data;Vectorizing data;Merging vectors.

#### 2.4.1. Aggregating Data

The aggregation step separates out the RSSI values of the moving device and aggregates them by time. The algorithm works on a per stationary device basis, so each stationary device produces one list of values. The list of RSSI values is sorted by timestamp, where many rows might have different data for similar timestamps. The output will be the same list, except each row will have a single data entry for each unique time.
1.Find all unique timestamps in the list.2.For each unique timestamp, t_s_:a.Find all RSSI values (${\mathrm{RSSI}}_{t_{s}}^{1}\ldots \;{\mathrm{RSSI}}_{t_{s}}^{n})$ for t_s_;b.Calculate mean of those RSSI values ${\mathrm{RSSI}}_{t_{s}}^{Avg}$;c.Create a new entry in the output for t_s_ with ${\mathrm{RSSI}}_{t_{s}}^{Avg}$.

For Algorithm 2, the attributes of the newly aggregated datasets consist of the following:Device name;Date;Time;RSSI (dBm) for S1;RSSI (dBm) for S2;RSSI (dBm) for S3.

#### 2.4.2. Combining Data

This step combines lists of RSSI values gathered from all three stationary devices. The prerequisite for this step is that the lists should already include the aggregated values. This algorithm is on a per mobile device basis. The input is the RSSI values sorted by timestamp, where each row has a single data entry for a unique time. In other words, the input for the combining step should be the output of the aggregate step. The expected output of this algorithm is a list of RSSI values sorted by timestamp, except each row has three RSSI values (for three stationary devices) corresponding to one unique timestamp.
For each row in the first stationary device’s time sorted RSSI data:a.Obtain all corresponding rows of data for other stationary devices;b.Calculate the average of the timestamps of all the stationary devices’ data for that row;c.Create a new row with the newly calculated timestamp and the RSSI values for each value;i.If there is no RSSI for a device for that row, place 0 for that device.

#### 2.4.3. Vectorizing Data

After combining the data, the algorithm vectorizes the list of RSSI values from a mobile device on a specific timestamp. The prerequisite for the vectorization step is that the data must be sorted by time. The input is the list of RSSI values from all three stationary devices, sorted by timestamp. The output for this step should be a path of vectors with starting and ending timestamps, and the source and the destination stationary devices. The vectors are represented with a tail (from stationary device) and a head (toward a stationary device) that indicate the movement of the mobile device. In the following algorithm, *P_st_* is used to represent the measured power received by the stationary device *s* at time t.
The source of the Path (*D_sr_*) is the stationary device that has the largest RSSI value at *t_start_*.The destination of the Path (*D_ds_*) is the stationary device that has the largest RSSI value at *t_end_*.For every *t_i_*:
a.Obtain the *i^th^* row values (RSSI values gathered at time t_i_) and designate that as *R;*b.Obtain the *j^th^ (i + 1)* row values (RSSI values gathered at time t_j_) and designate that as *S*;c.For every RSSI value in *R*:i.if *P_si_ < P_sj_*: This indicates that the device is moving away from the current location (stationary device *s*) toward the destination (*Ds*); then, create a vector *D_s_*
*→ D_ds_.*
ii.if *P_si_ > P_sj_*: This indicates that the device is moving closer to the source (*D_sr_*); then, create a vector *D_s_*
*→ D_sr_.*


#### 2.4.4. Merging Vectors

The final step involves merging a list of vectors to form a visualization of a path relative to the three stationary devices. For the input, this step requires the vectors from the previous step. The expected output is a sequence of stationary devices (S_a_, S_b_, …, S_n_), denoting a path indicating the movement of the mobile device from times *t_start_* to *t_end_*.
Assign *D_sr_* in the beginning of the path.For every vector:a.If the tail is equal to *D_sr_* and the head is equal to *D_ds_*:This is already in the path, so ignore.b.If the tail is equal to *D_sr_*, but the head is not equal to *D_ds_*:Append the head of the vector to the path.
c.If the tail is not equal to *D_sr_*, but the head is equal to *D_ds_*:
Append the tail of the vector to the path.
d.If the tail is not equal to *D_sr_*, and the head is not equal to *D_ds_*:
Append the tail of the vector to the path;Append the head of the vector to the path.Append *D_ds_* to the end of the path.

## 3. Results

This section presents the results from the two experiments and the path-forming algorithms. The resulting visualizations created from the algorithms are presented for the experiments. For estimating locations, an evaluation table containing the distances between predicted and actual locations is introduced. For our results on path estimation, the actual paths are presented along with the visualizations generated by the proposed algorithms to demonstrate the efficacy of the algorithms. 

### 3.1. Results for Estimating Locations

This section details the results of the experiment that estimates the location of an idle mobile device within the indoor space (Section 2.1.1). In that experiment, the mobile device was motionless at a single location, while the stationary devices around the indoor space obtained the signal strengths for a certain amount of time. 

Figure 4 shows the results of the estimated locations against the actual recorded locations (1–15) of the mobile device. The locations of the stationary devices in the indoor space are shown as blue center points in the figure along with the circles with RSSI values as their radiuses. Stationary Device 1 (S1) is plotted at {0, 0}, Stationary Device 2 (S2) is plotted at {258, 683}, and Stationary Device 3 (S3) is plotted at {531, 176}. The red points in the figure represent the points where the circles intersect. The orange point indicates the actual location of the mobile device, while the green point is the estimated location. The measured power was -44.0 dBm, and the constant “N” was set as 3.1 to calculate all distances using the previously described equation.

Locations 2, 8, 9, and 15 show instances where the estimated location was very close to the actual location. While not as close as the last four, Locations 6, 10, 11, 12, and 14 show the estimated and actual locations being in close proximity. The other six locations indicate a case where the estimated locations were not close.

### 3.2. Results for Estimating Paths

This section presents the results from estimating the paths taken during the second experiment (Section 2.1.2). This experiment involves gathering RSSI data at stationary devices while a volunteer takes the mobile device and traverses different paths in real time. This section also contains our estimation using manually generated simulated data. This allows for a comparison between real data and simulated data. 

The first algorithm applies the approach from the first phase of experiments, i.e., using the center of the trilateration area to find an approximate location. However, it extends that approach by combining these locations to form a visualization of a path. The second algorithm uses the data to create vectors indicating the device’s movement. These vectors form a sequence of stationary device proximities, indicating a path. This section shows not only the results of the algorithms but also the paths taken to show what the algorithms were supposed to predict.

#### 3.2.1. Estimation Using Simulated Data

The manually generated path from the simulated data consists of directly approaching and passing by each stationary device in a straight line. The sequence of the path is denoted as S1 -> S2 -> S3. Figure 5 shows a visualization of the layout:

Figure 6 presents Algorithm 1′s attempt at replicating the simulated path. In the plot, the beginning of the path is denoted by the black line. The color of the path increases in brightness, showing the progression of the path as it comes closer to its end. The visualization shows that the algorithm was able to successfully estimate the path from the simulated data. The starting point is located away from S1, and the end point is located away from S3 on the right side. However, the sequence of passed devices is correctly estimated.

For Algorithm 2, the algorithm shows a visualization with arrows from one stationary device to another. The numbers next to the arrows represent the number of vectors indicating movement away from one device and toward the other. For the simulated path in Figure 7, the visualization of Algorithm 2 shows three instances of direction between stationary devices and the number of vectors. 

The direction between devices and the count of vectors are found in sequential order in a separate output of Algorithm 2, as shown in Figure 8. After the counts of vectors for each direction, there is a sequence of stationary device proximities denoting a path. Here, the sequence is S1, S2, S3, which is correctly estimated.

The results from Algorithm 2 show that both algorithms were able to successfully replicate the path from the simulated data.

#### 3.2.2. Estimation of Path 1

The first path involved the volunteer moving from S1 to S3, S3 to S2, and S2 back to S1, as shown in Figure 3a. Figure 9 presents Algorithm 1′s attempt to replicate the path taken by the volunteer during the experiment. Looking at the plot, Algorithm 1 was able to replicate Path 1. There are some minor differences, such as the line segments from S1 to S3 having a zig-zag formation rather than a straight line, or the line segment from S2 moving toward the midpoint of S1 and S3. Regardless of the minor inconsistencies, Algorithm 1 was successful in estimating the general traversal of Path 1.

For Algorithm 2, the algorithm shows a visualization with arrows from one stationary device to another. In Figure 10, the visualization of Algorithm 2 shows four instances of direction between stationary devices and the number of vectors. 

The output from Figure 11 shows the counts of vectors for each direction and a sequence of stationary device proximities denoting a path. Here, the sequence is S1, S2, S3, S2, S1. This is very close to the sequence that the volunteer took during Path 1. The only difference in the output is that the sequence shows S2 before S3, which did not happen in Path 1.

#### 3.2.3. Estimation of Path 2

The second path involved the volunteer moving in a straight line from the left side to the right side of the indoor space, as shown in Figure 3b. The volunteer never directly approaches any of the stationary devices but passes them along the way. 

Figure 12 shows Algorithm 1′s estimation of Path 2, which is not exactly ideal. The first noticeable difference is that the path is not in a straight line. Although the starting and ending points are somewhat similar, the path in the plot immediately diverges and moves toward S2 before immediately approaching S3. If the paths were to end there, then this would not be a bad estimation from the algorithm. However, the plot shows the path moving to the midpoint of S1 and S3 before ending back on the right side of the layout.

For Algorithm 2, the sequence of stationary device proximities shown in Figure 13 is S1, S2, S3, S2, S3, S1. The first half of the sequence correctly identifies the devices passed in Path 2. However, Algorithm 2 makes the same mistake as Algorithm 1 and diverges from the initial path. As noted by the output (Figure 14), the algorithm assumes that the volunteer moved back to S2 and returned to S3 again before moving to S1 at the end.

Unlike Path 1, the algorithms were not as accurate in estimating the correct flow of movement for Path 2 from start to end. Initially, both algorithms were able to estimate the right flow of movement, but they diverged to other directions, thus making their estimation less accurate.

#### 3.2.4. Estimation of Path 3

The third path involved the volunteer approaching the stationary devices in ascending order, from S1 to S2, S2 to S3, and S3 back to S1, as shown in Figure 3c. In Figure 15, the plot presents Algorithm 1′s attempt to replicate Path 3. The major difference of this visualization is the reversal of the order of stationary devices. Rather than showing movement from S1 to S2, the algorithm estimated movement from S1 to S3. Furthermore, it predicted movement toward S2 after S3 before returning to S1. The algorithm correctly identified the start and end points as S1 but flipped the sequence between S2 and S3.

For Algorithm 2, there are four instances of direction between stationary devices and the number of vectors indicating the movement of each direction. This is shown in Figure 16.

Looking at the output of Algorithm 2 (Figure 17), the sequence of stationary device proximities is S1, S2, S3, S2, S3, S1. Once again, Algorithm 2 makes the mistake of initially estimating the right path before diverging toward other devices. The first half of the sequence correctly identifies the right order of devices passed in Path 3. However, the algorithm assumes that the volunteer moved back to S2 and returned to S3 before moving to S1 at the end.

For Path 3, Algorithm 2 estimated the flow of movement more accurately than Algorithm 1. Both algorithms correctly identified the start and end points as S1. Algorithm 1 flipped the sequence from S2 to S3, to S3 to S2. The first half of Algorithm 2′s sequence was correct, but it also estimated that the volunteer went back to S2 and S3 twice before returning to S1.

## 4. Discussion

Table 2 shows the evaluation of the distance from the estimation of the algorithm to the actual coordinates on the graph. Location 9 seems to have the most accurate estimation, as the distance between the actual coordinates was 52.8 inches (4.4 feet). Nine locations (Locations 2, 6, 8, 9, 10, 11, 12, 14, and 15) had estimates that were less than 20 feet away. When looking at the figures for these nine locations, the estimated locations are in close proximity to the actual locations. As for the other six locations, these figures indicate that the estimates were not close.

With the simulated data, both algorithms were able to successfully estimate the sequence of stationary devices passed by. When making estimates from real data, the algorithms show potential in estimating paths, but there were a few discrepancies in the estimations. This is shown in Table 3, where the accuracy of the simulated data was perfect for both algorithms but performed worse on real data.

The estimations from the simulated data present an ideal case, where the RSSI value is scaled depending on the proximity between devices. This case assumes that there is no outside interference from any noise, such as people, obstacles, or other nearby devices connected to Bluetooth. On the other hand, the estimations on real data are not as accurate because of the data being affected by noise. From these experiments, it can be seen that the algorithms perform well in situations where there is little to no noise. However, their performance suffers in cases where noise is very significant, i.e., RSSI data collected in a public environment.

In the future, one of the first steps to emphasize in this research is to further optimize and tune the algorithms. For data collection, there are two challenges that need to be investigated. First, there is a need to improve the detection of RSSI data in noisy environments, as the method is very sensitive to noise. All noise has a negative effect on RSSI data collection, as it can cause variation within the dataset. In the experiments, the interference from the noise caused our RSSI values to fluctuate heavily, regardless of the mobile device’s positioning in the indoor space. The second challenge to be investigated is the synchronization of the stationary devices’ data collection. All stationary devices sensed the RSSI values of the mobile devices asynchronously because of how the app in the stationary devices is scheduled by the operating system. As a result, there were delays in data collection. For example, one stationary device may start collecting RSSI values immediately after detection, but the other devices may take longer to start. Times are not consistent among all stationary devices, and further investigation is needed to identify new effects stemming from synchronized data collection.

## 5. Conclusions

There are many causes of waste in the United States healthcare system, one of which is inefficiency. Inefficiency is a considerable concern in pro bono healthcare clinics. A proposed solution to mitigate this problem is clinic layout optimization, where optimizing the movement of practitioners and the layout of the indoor space can increase time with patients and improve patient care. Therefore, this research aimed to study and analyze methods for automatically identifying mobility patterns using Bluetooth RSSI data.

Toward this goal, experiments were conducted to gather indoor location data, and two path-forming algorithms were developed to estimate mobility in the indoor space. The first path-forming algorithm converts the RSSI data at each timestamp to a radius for each stationary device. The center of the trilateration area is the predicted location. The distance results between the estimated and the actual location show promise in estimating a device’s location. The second algorithm used the data to create vectors indicating the mobile device’s movement away from one stationary device to another. After vector creation, these vectors formed a sequence of stationary device proximities indicating a path. The results from estimating locations demonstrate that the algorithms can predict the general flow of movement based on the traversal through stationary devices.

Overall, this research concludes that there is potential in modeling movement automatically in an indoor space as well as estimating and visualizing a general path by using just the RSSI data.

## Figures and Tables

**Figure 1 sensors-22-07546-f001:**
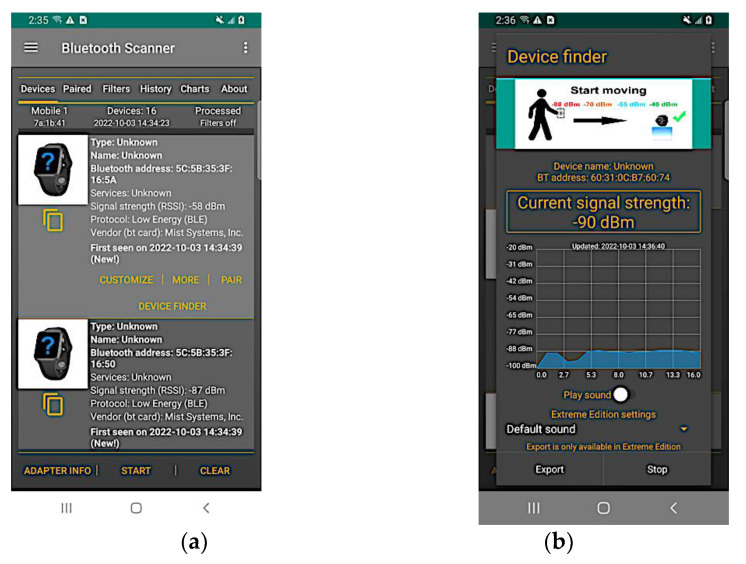
Data and visualizations provided by the Bluetooth Scanner app. (**a**) List of devices scanned by the app; (**b**) Bluetooth signal strength of a device gathered by the app.

**Figure 2 sensors-22-07546-f002:**
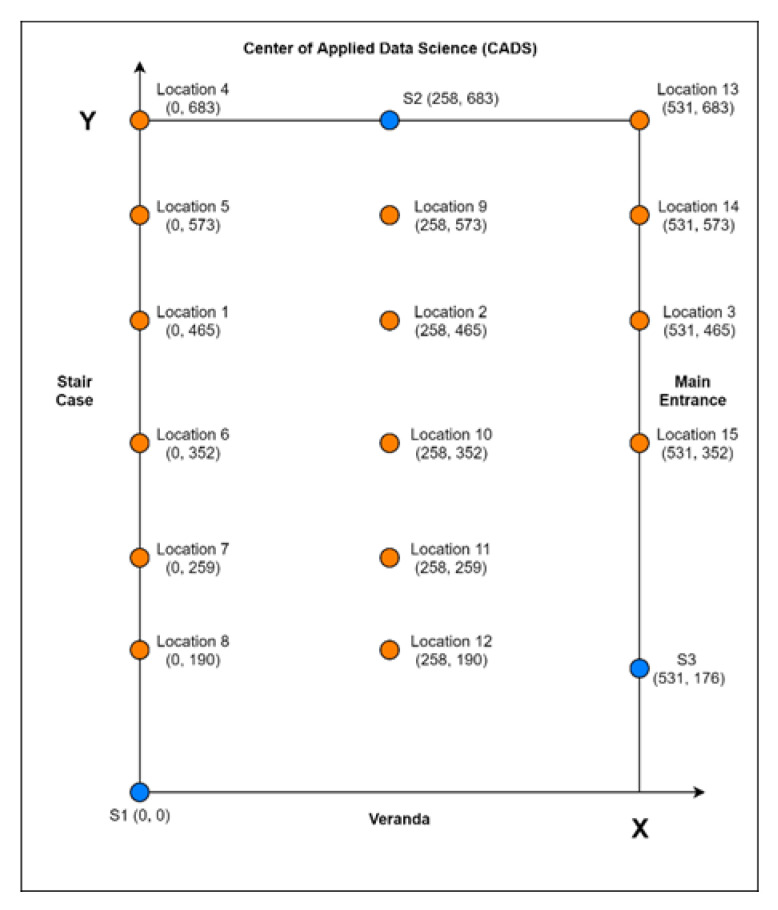
Layout of the indoor space with placements of devices.

**Figure 3 sensors-22-07546-f003:**
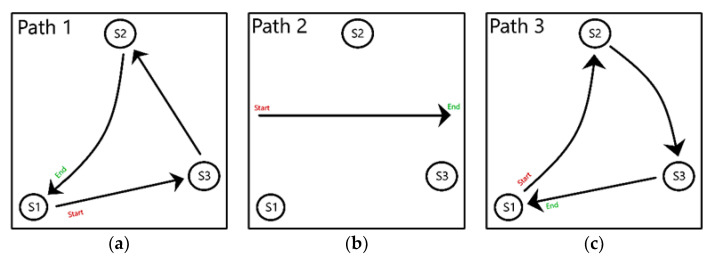
Different actual paths taken by the mobile device in the indoor space. (**a**) Path from S1 back to S1 through S3 and S2; (**b**) Path between the stationary devices; (**c**) Path from S1 back to S1 through S2 and S3.

**Figure 4 sensors-22-07546-f004:**
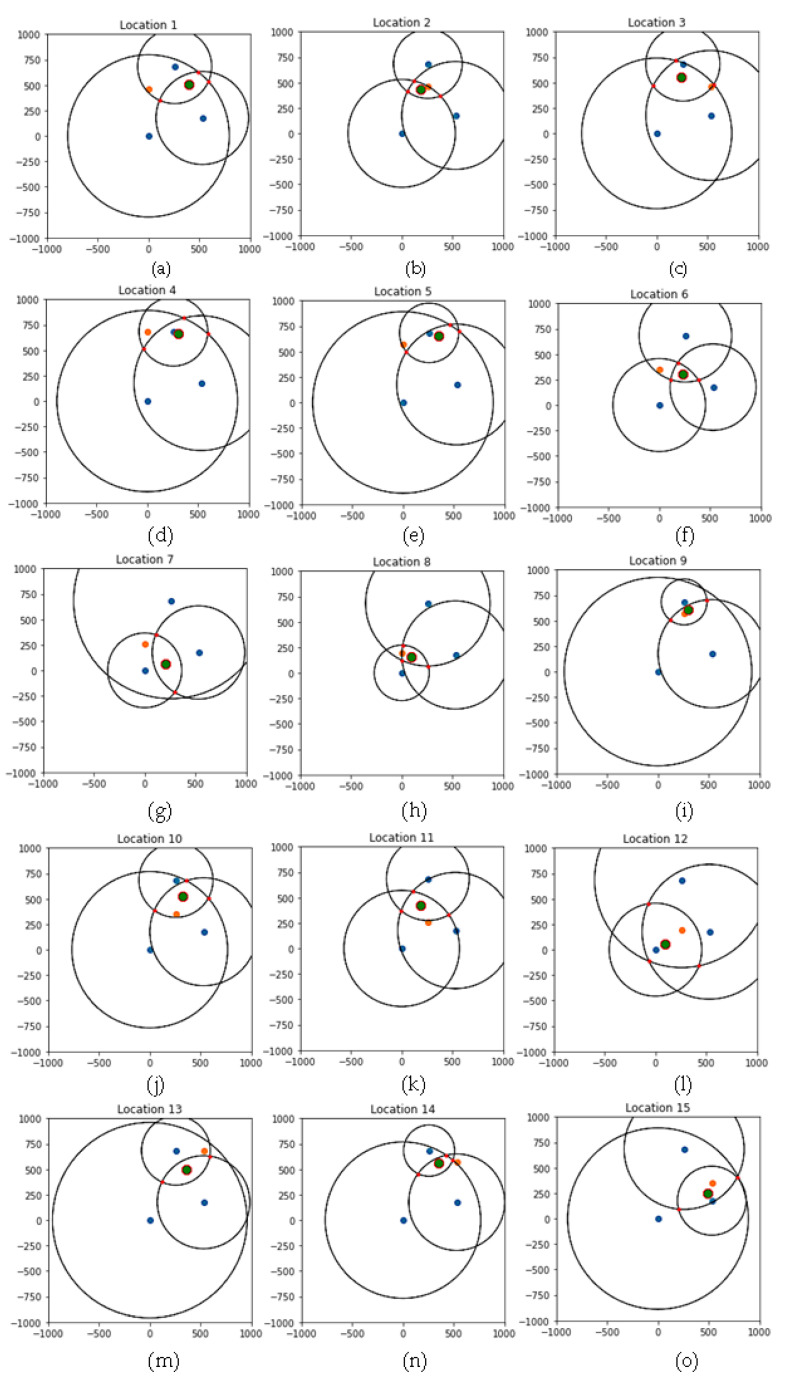
Estimated locations for each fixed location. (**a**) Estimated location for location 1; (**b**) Estimated location for location 2; (**c**) Estimated location for location 3; (**d**) Estimated location for location 4; (**e**) Estimated location for location 5; (**f**) Estimated location for location 6; (**g**) Estimated location for location 7; (**h**) Estimated location for location 8; (**i**) Estimated location for location 9; (**j**) Estimated location for location 10; (**k**) Estimated location for location 11; (**l**) Estimated location for location 12; (**m**) Estimated location for location 13; (**n**) Estimated location for location 14; (**o**) Estimated location for location 15.

**Figure 5 sensors-22-07546-f005:**
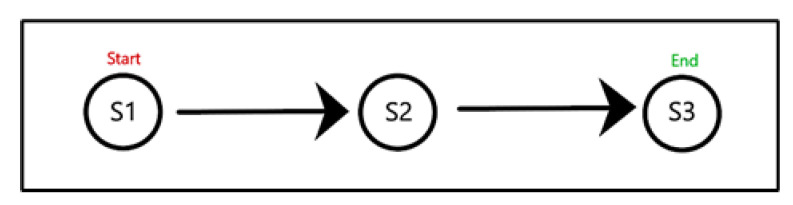
Layout of the simulated path in a straight line.

**Figure 6 sensors-22-07546-f006:**
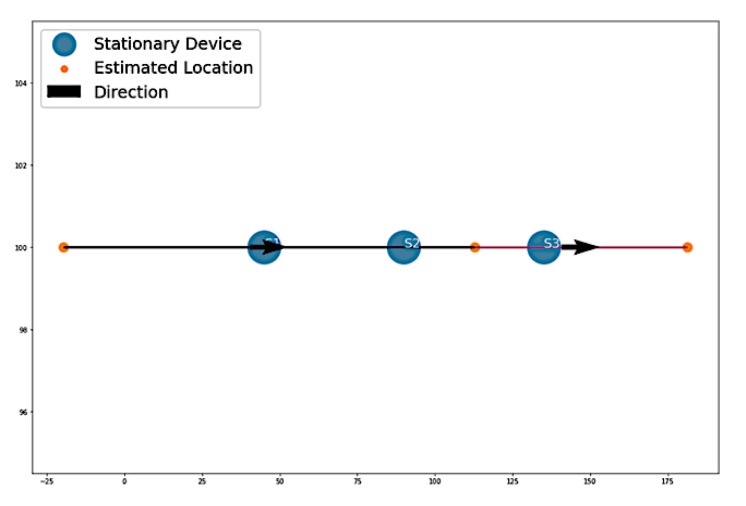
Algorithm 1’s estimation of the simulated path.

**Figure 7 sensors-22-07546-f007:**
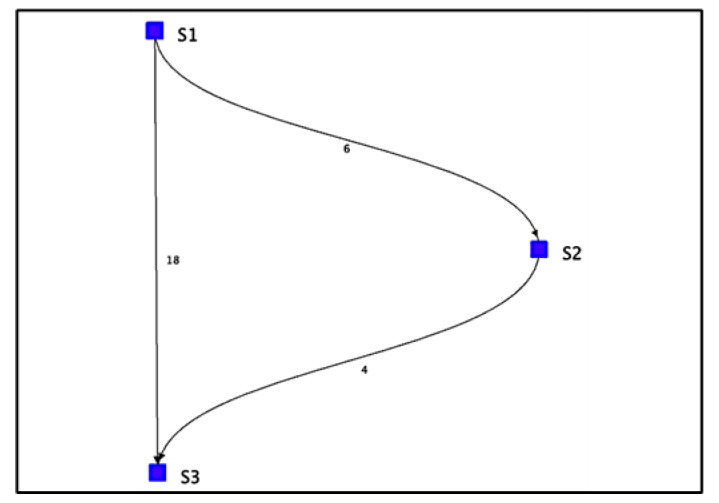
Algorithm 2’s visualization of simulated path vectors.

**Figure 8 sensors-22-07546-f008:**
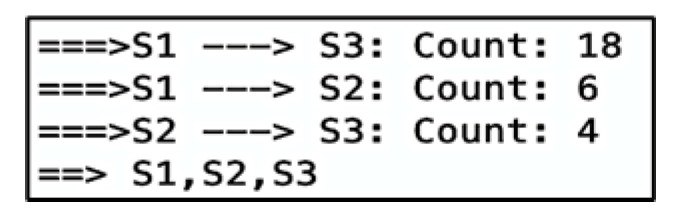
Algorithm 2’s output containing the paths and the count of vectors.

**Figure 9 sensors-22-07546-f009:**
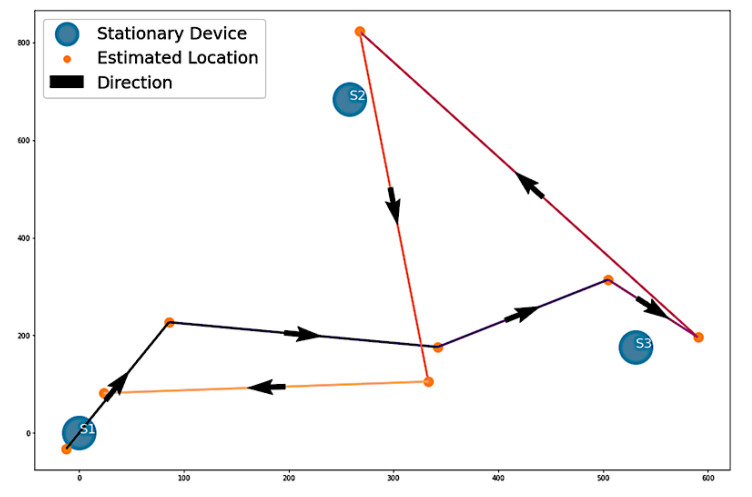
Algorithm 1’s estimation of Path 1.

**Figure 10 sensors-22-07546-f010:**
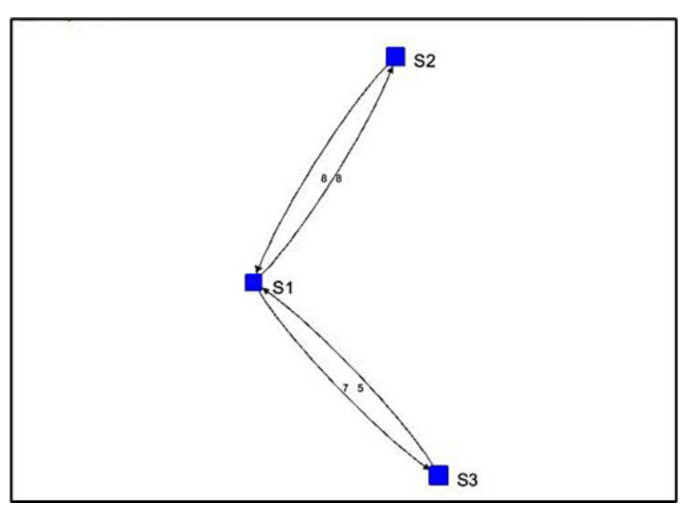
Algorithm 2’s visualization of Path 1 vectors.

**Figure 11 sensors-22-07546-f011:**
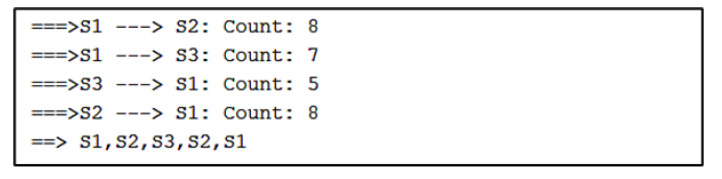
Algorithm 2’s output for Path 1.

**Figure 12 sensors-22-07546-f012:**
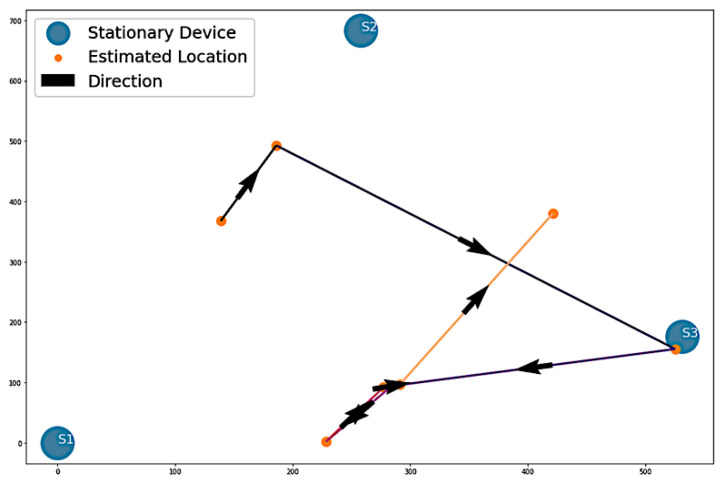
Algorithm 1’s estimation of Path 2.

**Figure 13 sensors-22-07546-f013:**
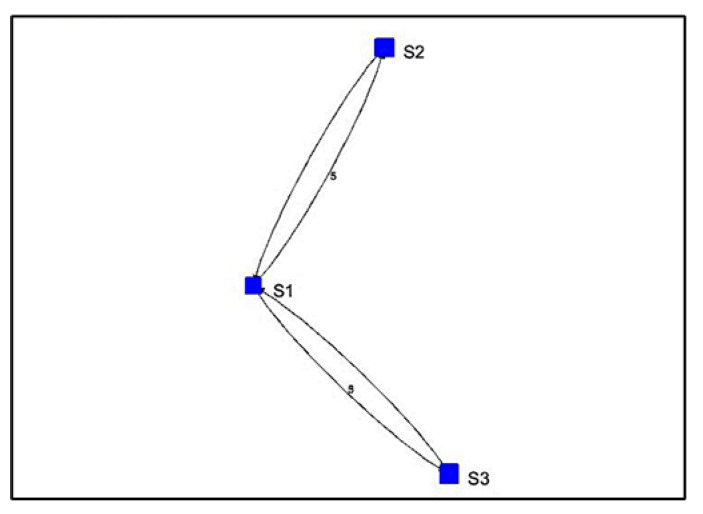
Algorithm 2’s visualization of Path 2 vectors.

**Figure 14 sensors-22-07546-f014:**
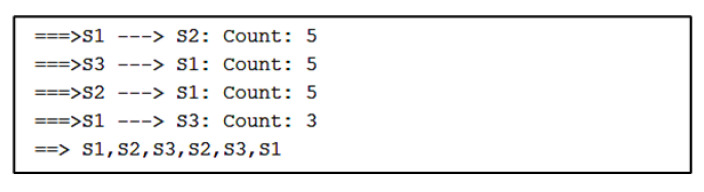
Algorithm 2’s output for Path 2.

**Figure 15 sensors-22-07546-f015:**
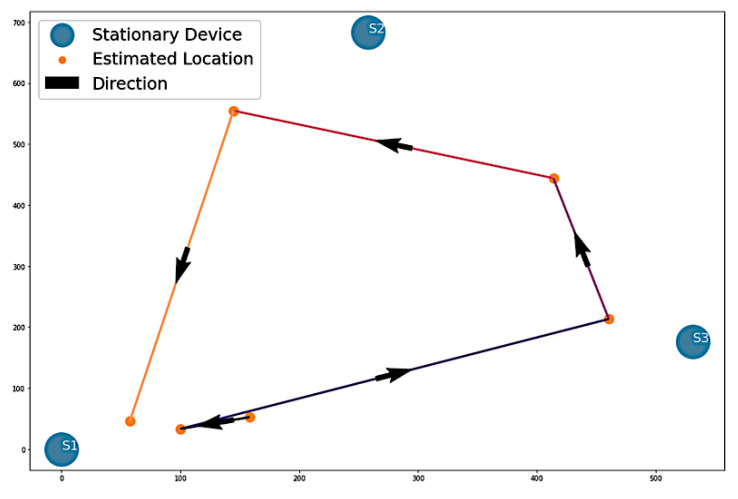
Algorithm 1’s estimation of Path 3.

**Figure 16 sensors-22-07546-f016:**
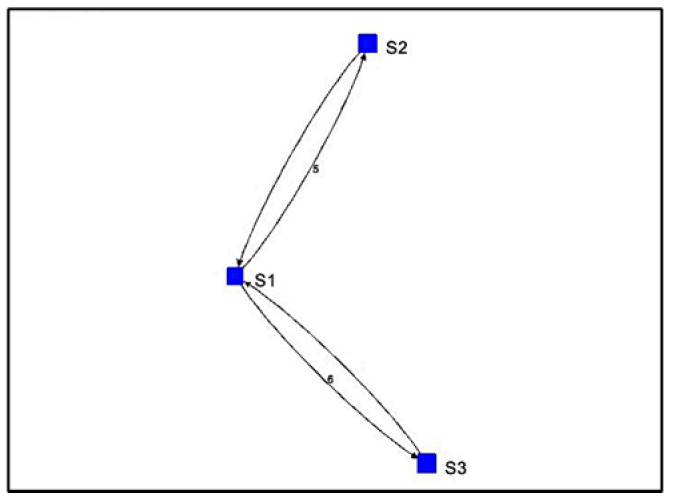
Algorithm 2’s visualization of Path 3 vectors.

**Figure 17 sensors-22-07546-f017:**
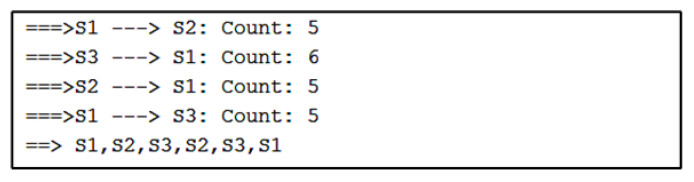
Algorithm 2’s output for Path 3.

**Table 1 sensors-22-07546-t001:** Similarities to and differences from related research.

Related Research	Similarities to Related Research	Differences to Related Research
[5]	None	Raw positioning data are used in this research No need to transform data into semantics in this research
[6]	Both use nodes and stationary devices Both make use of vectors	This research does not utilize query processing for data
[10]	Both used stationary and moving nodes	This research using a dedicated mobile application to receive RSSI data
[12]	None	This research uses mobile devices instead of IoT sensors This research uses provider data
[13]	Both show visualization of movement in a specific location	The related research only used heatmaps to show quantity of movement
[14]	Both focused on mobile crowdsensing	This research uses RSSI data This research does not utilize query processing This research does not use a toolbox for data retrieval

**Table 2 sensors-22-07546-t002:** Table containing distances from estimation to actual location.

	Estimated Coordinates	Actual Coordinates	Distance (inches)	Distance (feet)
Location 1	(0, 465)	(396, 504.8)	397.9	33.2
Location 2	(258, 465)	(187.3, 430.9)	79.2	6.6
Location 3	(531, 465)	(238.7, 554.4)	306.2	25.1
Location 4	(0, 683)	(306.3, 664.5)	306.6	25.6
Location 5	(0, 573)	(347.2, 653.9)	356.3	29.7
Location 6	(0, 352)	(227.4, 304.2)	232	19.3
Location 7	(0, 259)	(201.9, 66.9)	279.3	23.3
Location 8	(0, 190)	(92.1, 154.3)	98.8	8.2
Location 9	(258, 573)	(299.8, 605.3)	52.8	4.4
Location 10	(258, 352)	(327.9, 523.5)	185.2	15.4
Location 11	(258, 259)	(189.5, 420.6)	175.5	14.6
Location 12	(258, 190)	(97.4, 59.9)	206.7	17.2
Location 13	(531, 683)	(356, 501)	252.5	21
Location 14	(531, 573)	(356, 561.4)	175.4	14.6
Location 15	(531, 352)	(491.5, 249.3)	110	9.2

**Table 3 sensors-22-07546-t003:** Comparison of algorithms’ estimated sequences.

Path	Original Sequence	Algorithm 1 Sequence	Algorithm 2 Sequence	Algorithm 1 Accuracy	Algorithm 2 Accuracy
Simulated Path	S1,S2,S3	S1,S2,S3	S1,S2,S3	100%	100%
Path 1	S1,S3,S2,S1	S1,S3,S2,S1	S1,S2,S3,S2,S1	100%	80%
Path 2	S1,S2,S3	S1,S2,S3, S1,S3	S1,S2,S3,S2,S3,S1	60%	50%
Path 3	S1,S2,S3,S1	S1,S3,S2,S1	S1,S2,S3,S2,S3,S1	50%	66%

## Data Availability

Not applicable.

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
