# Peer review of "Path Forming of Healthcare Practitioners in an Indoor Space Using Mobile Crowdsensing"

_sensors, 2022, doi:10.3390/s22197546_

Round 1
Reviewer 1 Report
The topic of the paper is interesting and up to date.
Please add the equation number to the equation in line 314 and after that please make a reference with the number in the line 328.
Author Response
Please see the attached document for a point-by-point explanation.

Reviewer 2 Report
It is very difficult to read this article. It should be well organized. The problem statement and experiments are confusing. I cannot catch the contributions. I feel I am reading an experiment report. The paper should be rewritten and well organized. I suggest the authors should clearly intuitively list the contributions at the end of the introduction, and use a paragraph to summarize the method at the beginning of the method. Based on the current version, I cannot suggest an Accept decision. BTW, there are a lot of related methods. Why did you select Ref [3-8] for comparisons in Table 1?
Author Response

(The authors gave the same response as above.)

Reviewer 3 Report
The abstract says that one of the solutions to address inefficiency is clinic layout optimization, which makes me be interested in this research. However, in the main content, such things disappeared.
There are some unclear and incoherent statements, such as Line 64-66, Line 74-76, Line 81-82, and Line 90-91.
After reading the introduction, I cannot picture the motivations of this research, especially what are the issues of the current research. Meanwhile, the things about path forming are missing.
The structure of Section 2 is not reasonable. My suggestion is that separate the methods from experiments. Moreover, the preconditional settings should be presented before experiments.
Figure 2. Why such a layout of indoor space with nothing special is selected? What are the orange and blue nodes? Why they are distributed in such a way? Although some explanations are included, I still cannot link them. In other words, the caption should be extended.
Only three paths are employed to illustrate the method, which is not that convincing. I mean the experiments in this manuscript are too weak. Please also refer to the last comment.
The caption of Figure 4 also should be extended with more explanations.
Line 858-871 should be moved in future work.
Author Response

(The authors gave the same response as above.)

Round 2
Reviewer 2 Report
This article has been reviewed before. However, the quality of the response letter is terrible. It may be the most simplest letter I read. I kindly suggest the authors consult more experienced researchers to improve their letter.
Author Response
This article has been reviewed before. However, the quality of the response letter is terrible. It may be the most simplest letter I read. I kindly suggest the authors consult more experienced researchers to improve their letter.
==> The authors are confused about this review, which doesn’t help them improve the paper. We individually answered each concern in the first review and updated the paper accordingly. We kindly ask you to let us know what was missing in that response that doesn’t answer the concerns raised in the review.
Reviewer 3 Report
I am satisfied with the replies and revised version.
Only two suggestions:
1. It is slightly difficult for reviewers to read a manuscript with revisions mode.
2. Please remove the background that this research can help avoiding wasting resources, since such things are mentioned in the title and abstract, however, nothing has been done related to this in the contents.
Author Response
- It is slightly difficult for reviewers to read a manuscript with revisions mode.
==> The authors were instructed by the journal to have all revisions in the document be shown.
- Please remove the background that this research can help avoiding wasting resources, since such things are mentioned in the title and abstract, however, nothing has been done related to this in the contents.
==> The authors believe that such context is necessary to understand this research's benefits and applicability. Therefore, we just mention that as a precursor to introducing this research.